# Fuzzy Synchronization of Chaotic Systems with Hidden Attractors

**DOI:** 10.3390/e25030495

**Published:** 2023-03-13

**Authors:** Jessica Zaqueros-Martinez, Gustavo Rodriguez-Gomez, Esteban Tlelo-Cuautle, Felipe Orihuela-Espina

**Affiliations:** 1Department of Computer Science, Instituto Nacional de Astrofísica, Óptica y Electrónica (INAOE), Tonantzintla, Puebla 72840, Mexico; 2Department of Electronics, Instituto Nacional de Astrofísica, Óptica y Electrónica (INAOE), Tonantzintla, Puebla 72840, Mexico; 3School of Computer Science, University of Birmingham, Edgbaston, Birmingham B15 2TT, UK

**Keywords:** synchronization, chaos, hidden attractor, fuzzy control, T–S fuzzy model

## Abstract

Chaotic systems are hard to synchronize, and no general solution exists. The presence of hidden attractors makes finding a solution particularly elusive. Successful synchronization critically depends on the control strategy, which must be carefully chosen considering system features such as the presence of hidden attractors. We studied the feasibility of fuzzy control for synchronizing chaotic systems with hidden attractors and employed a special numerical integration method that takes advantage of the oscillatory characteristic of chaotic systems. We hypothesized that fuzzy synchronization and the chosen numerical integration method can successfully deal with this case of synchronization. We tested two synchronization schemes: complete synchronization, which leverages linearization, and projective synchronization, capitalizing on parallel distributed compensation (PDC). We applied the proposal to a set of known chaotic systems of integer order with hidden attractors. Our results indicated that fuzzy control strategies combined with the special numerical integration method are effective tools to synchronize chaotic systems with hidden attractors. In addition, for projective synchronization, we propose a new strategy to optimize error convergence. Furthermore, we tested and compared different Takagi–Sugeno (T–S) fuzzy models obtained by tensor product (TP) model transformation. We found an effect of the fuzzy model of the chaotic system on the synchronization performance.

## 1. Introduction

Synchronization of chaotic systems has received substantial attention theoretically and experimentally in recent years. Such attention is justified by the potential applications in secure communication [1,2,3], the control of chemical reactions with the aim of determining the favorable conditions for practical implementation [4], the stability of the chaotic wind power system in finite time [5], the synchronization of chaotic finance systems with known and unknown systems parameters [6,7], authentication from brain signals [8], chaotic attitude synchronization and anti-synchronization for master–slave satellites under unknown moments of inertia and disturbance torques [9], regulation of glucose–insulin concentrations from a chaotic regime (an illness) to a disorder-free equilibrium [10], and the relation between a meteorological phenomenon and influenza pandemics [11].

The synchronization of dynamic systems is the problem of enforcing several systems to match and converge [12]. Chaotic systems are deterministic nonlinear dynamic systems that exhibit a prominent sensitivity to initial conditions, noise, and system parameters [13]. Such characteristics of chaotic systems make the problem of synchronizing these systems particularly challenging. In the case of chaotic systems, it is known that a chaotic solution is neither asymptotically stable, nor totally unstable, but it alternates between two or more attractors over time.

Moreover, the synchronization of chaotic systems with hidden attractors is even more challenging as they can lead to erroneous coupling behavior and instability. Besides, the system analysis can be complicated [14,15,16]; on the one hand, hidden attractors have been observed in systems without no equilibrium points, with no unstable equilibrium points, with one stable equilibrium point, or with a line equilibrium. On the other hand, the basins of attraction of the hidden attractors do not intersect with small neighborhoods of any equilibrium points (if any) and are located far from such points. To avoid or reduce the risk of a sudden jump to unwanted behavior, it is helpful to know the properties of hidden attractors.

In order to carry out the synchronizations, it is necessary to leverage numerical systems so as to approximate the solutions of the systems involved. The global error $e_{n}\left(t\right)=y_{n}-y\left(t_{n}\right)$, where $y_{n}$ is the approximation to the theoretical value $y(t_{n})$ at time $t_{n}$, of the numerical method is upper bounded by the error made on the initial conditions, the truncation error, and the rounding-up error [17,18]. The inadequate choice of the integration step may result in the growing of the truncation error and/or the rounding error. Consequently, the system may be perturbed, yielding undesired behaviors or even making the system inoperable in the presence of hidden attractors.

The goal of this research is to afford a new methodology to carry out the synchronization of two identical chaotic systems with hidden attractors by means of fuzzy control and a special integration method that exploits the oscillating property of the chaotic systems for numerical integration.

Based on our previous expertise, we opted for complete and projective synchronization schemes. The complete synchronization employs the linear matrix inequality (LMI), whilst the projective synchronization employs parallel distributed compensation (PDC). The former proposal was applied to a set of chaotic systems with hidden attractors known in the literature.

The main contribution of this research is to provide a new methodology to synchronize chaotic systems with hidden attractors through complete and projective fuzzy controls, evidencing that those fuzzy control strategies combined with such special integration method boost robustness over other popular approaches and, hence, are more likely successful in the task of synchronizing chaotic systems with hidden attractors.

The convergence to zero of the error in the synchronization varies depending on the strategy used for the control. Projective synchronization depends on two parameters, α and γ. We further propose here a new strategy for selecting the control parameter γ, then we manipulated the scaling parameter α, as well as the initial conditions to modulate the speed of convergence according to the demands of the application. For this, we supported our findings with numerical simulations.

Another contribution is to evidence that fuzzy controls used in this work are robust in handling multistable systems with hidden attractors, and the fuzzy controls used are also robust to changes in the initial conditions. Another contribution is that the tested chaotic systems were chosen arbitrarily from the literature. Therefore, these systems are interchangeable with any other continuous integer-order chaotic system with nonlinear terms with a state variable in common. Few works in the literature emphasize the numerical methods used to simulate chaotic systems since they can generate problems such as superstability. Our proposal is to use a special fixed-step numerical method that exploits the oscillatory characteristic of chaotic systems and where the integration step is easy to choose. Furthermore, an analysis of the performance of the synchronizations using different Takagi–Sugeno (T–S) fuzzy models was conducted. The results showed an effect of the fuzzy model of the chaotic system on the synchronization performance.

## 2. Related Work

There are two configurations in synchronization: bidirectional or mutual coupling systems, where both systems must adapt to each other, and master–slave systems or unidirectional coupling in which one system must adapt to another [19,20]. In both configurations, the synchronization efforts can be categorized by the type of error being minimized [12], such as complete or identical synchronization, anti-synchronization, phase synchronization, and projective synchronization, among other synchronizations.

The literature reports several efforts regarding the synchronization of chaotic systems in the presence of hidden attractors. One of the most-used controls is feedback control with complete error [16,21,22,23], synchronizing with one or two state variables with the master–slave and bidirectional approach [20,24]. To use this control, it is necessary to adjust the coupling parameter to achieve synchronization. This can be done through bifurcation diagrams, but they are very computationally expensive. This control intentionally fails when synchronizing Chua’s system with hidden attractors to show that the existence of hidden attractors significantly affects synchronization [15]. Another control is sliding mode control (SMC) with complete error or anti-synchronization [25,26,27,28]. However, this control requires studying and understanding the theory, which has strong mathematical foundations. Most of the time, researchers focus on solving the problem of synchronizing systems with perturbations and unknown system parameters. Another used control is adaptive control with complete error or anti-synchronization, where they focus on the time of convergence of the error in the synchronization [29] or the problem where the system parameters are unknown [30,31,32,33,34]. Another control proposal is the *fixed-time synchronization observer*, which emphasizes the time of convergence of the error in the synchronization under complete error [35]. Finally, another control, *back-stepping control*, synchronizes two different systems with complete error by setting an extra variable [36].

Most of the controls reviewed are specific regarding the set of chaotic systems that they synchronize. For example, SMC works with perturbed systems and with unknown parameters. Other controls are interested in the analysis of the convergence time of the synchronization. In this article, we addressed the problem of synchronizing two identical chaotic systems with hidden attractors considering the complete and projective errors. To solve this problem, it is necessary to have a control, but the choice of this control is not straightforward in the presence of hidden attractors [16]. In addition, a general control was sought that guarantees the solution to a broad range of chaotic systems, avoiding ad hoc controls, which particularize the solution.

On the other hand, fuzzy control is an efficient and effective tool in nonlinear systems and has been successfully applied to chaotic synchronization before [37]. Fuzzy control theory is widely studied. Furthermore, the part that is used to perform the synchronization does not require a priori knowledge of fuzzy control, nor specific skills to use it [38]. Fuzzy control applied to the synchronization of chaotic systems is a general and unified control because it synchronizes a variety of chaotic systems, unlike controls such as linear feedback control, nonlinear feedback control, and impulsive control, which deal with one or two kinds of specified chaotic systems [37].

## 3. Global Error

A known result in the literature is how the global error $e_{n}\left(t\right)$ of the numerical approximation of a solution of ordinary differential equations with initial values is bounded according to
$$|e_{n}\left(t\right)|<\left\{A\delta +Bh^{p}+Ch^{q}\right\}$$
where A,B,C are constants that depend on the methods’ coefficients and the Lipschitz constant of ordinary differential equations and δ is the bound initial conditions [17,18].

The bounds of the global error are then given in terms of the error in the initial values δ, the discretization or truncation error, and the round-off error. The choice of the integration step ought to maintain a compromise between the later two. Besides, the error in the initial values in the floating point arithmetic ought to be of the same order as that of the integration method. Because of the sensitivity to the initial conditions, in the case of dynamic chaotic systems, the integration step *h* and the error in the representation are of utmost importance. If we choose an integration step that keeps the discretization error bounded, but which does not bound the growing of the round-off error, the later can alter the chaotic system; for example, the chaos may be lost (*superstability*) or make the system attracted to a different attractor, thus changing the system’s behavioral regime. Even the variable step numerical integration methods with error control exhibit difficulties when choosing the integration step to constrain the growth of the error e(t), leading to the so-called *computational chaos* of systems that are not originally chaotic. For example, Skufca reported how the Runge–Kutta–Fehlberg method implemented with error control and a variable step (ode45 in Matlab) when applied to a system that models the coupling of two oscillators originates the computational chaos [39].

## 4. Special Numerical Methods

Whenever some characteristic or property of the dynamical systems is known, said property can be exploited by the numerical method. Special numerical methods are those that exploit some specific property present in the dynamical system. In such circumstances, the numerical approximation to the solution of the ordinary differential equations with some initial values can be enhanced capitalizing on such prior knowledge about the dynamical system. It follows that, since chaotic systems exhibit oscillations, it is convenient when integrating them to use a method that leverages such an oscillatory property. Therefore, in this research, we opted for using the numerical method developed by Gautschi [40]. This is a well-known method based on trigonometric polynomials that effectively exploits this oscillatory property.

Gautschi’s method has been described and analyzed in detail in [17,40], but briefly, it is an explicit method of trigonometric order q=1 and algebraic order p=2, given by
$$y_{n+2}-y_{n+1}=h\left(\beta_{1}f_{n+1}+\beta_{0}f_{n}\right)$$
where
$$\begin{array}{ccc} \beta_{0}\left(\nu \right) & = & -\frac{1}{2}\left(1+\frac{1}{12}\nu^{2}+\frac{1}{120}\nu^{4}+\ldots \right)\\ \beta_{1}\left(\nu \right) & = & \frac{3}{2}\left(1-\frac{1}{4}\nu^{2}+\frac{1}{120}\nu^{4}+\ldots \right) \end{array}$$
with ν=ωh, the integration step *h* constant, and the truncation error $\frac{5}{12}h^{3}\left(y^{\left(3\right)}+\omega^{2}y^{\left(1\right)}\right)$. If the frequency ω is unknown, it is safer to underestimate it than to overestimate it [17].

## 5. Fuzzy Control

Fuzzification translates crisp inputs into fuzzy values for the inference system. The fuzzy inference system applies the fuzzy rules and generates the system outputs in fuzzy form. The counterpart, defuzzification, proceeds to recover crisp values for the outputs.

### 5.1. Fuzzification

Here, the fuzzy inference system employs the T–S fuzzy dynamic model [41]. The T–S model is described by a set of rules: *IF*–*THEN*. These rules are capable of exactly describing nonlinear systems within some region of interest. Nonlinearities are expressed as inferred fuzzy outputs by indicating the fuzzy membership function in the premise and the associated coefficients in the consequence [42].

Consider the continuous-time chaotic system given by
(1)$$\dot{x}=f\left(x\left(t\right)\right)$$
where $x\left(t\right)={\left[x_{1}\left(t\right),x_{2}\left(t\right),\ldots ,x_{n}\left(t\right)\right]}^{T}\in R^{n}$ is the state vector and f is a nonlinear function of appropriate dimensions. The T–S fuzzy model is composed of a set of rules of the form:
(2)$$\begin{array}{cc} \mathrm{Rule}\;i: & \mathrm{IF}\;z_{1}\left(t\right)\;\mathrm{is}\;F_{i1}\mathrm{and}\ldots \;\mathrm{and}z_{p}\left(t\right)\;\mathrm{is}\;F_{ip}\\ & \mathrm{THEN}\;\dot{x}=A_{i}x\left(t\right)+b_{i},\;i=1,2,3,\ldots ,I \end{array}$$
where $z_{1}\left(t\right)\ldots z_{p}\left(t\right)$ are the premise state variables, $F_{i,j}$ (j=1,…,p) are fuzzy sets, $A_{i}$ is a constant system matrix with the appropriate size, and $b_{i}\in R^{n}$ is a constant vector; it represents the bias generated by the exact fuzzy model. The number of rules *I* depends on the variables exhibiting the nonlinearities.

### 5.2. Defuzzification

Using the singleton fuzzifier, fuzzy inference by the product, and defuzzification by the weighted average, the final outputs are calculated according to
(3)$$\dot{x}=\frac{\sum_{i=1}^{I}\omega_{i}\left(z\left(t\right)\right)\left\{A_{i}x\left(t\right)+b_{i}\right\}}{\sum_{i=1}^{I}\omega_{i}\left(z\left(t\right)\right)}$$
where $z\left(t\right)={\left[z_{1}\left(t\right)\;z_{2}\left(t\right)\;\ldots \;z_{p}\left(t\right)\right]}^{T},$ with $w_{i}\left(z\left(t\right)\right)=\prod_{j=1}^{p}F_{ij}\left(z_{j}\left(t\right)\right)$ in which $F_{ij}\left(z_{j}\left(t\right)\right)$ is the degree of membership of $z_{j}\left(t\right)$ in $F_{ij}$, with $\sum_{i=1}^{I}\omega_{i}\left(z\left(t\right)\right)>0$, and $\omega_{i}\left(z\left(t\right)\right)\geq 0$, i=1,2,3,…,I.

By employing $\mu_{i}\left(z\left(t\right)\right)=\omega_{i}\left(z\left(t\right)\right)/\sum_{i=1}^{I}\omega_{i}\left(z\left(t\right)\right)$ instead of $\omega_{i}\left(z\left(t\right)\right)$, System (Equation 3) is rewritten as
(4)$$\dot{x}=\sum_{i=1}^{I}\mu_{i}\left(z\left(t\right)\right)\left\{A_{i}x\left(t\right)+b_{i}\right\}$$
where $\sum_{i=1}^{I}\mu_{i}\left(z\left(t\right)\right)=1$ with $\mu_{i}\left(z\left(t\right)\right)\geq 0$ (i=1,2,…,I).

When modeling a chaotic system using a T–S fuzzy surrogate, the goal is to build a T–S model following (Equation 2), representing exactly the nonlinear system in (Equation 1). The vector function f(x(t)) is expressed as the inferred fuzzy output $\sum_{i=1}^{I}\mu_{i}\left(z\left(t\right)\right)\left\{A_{i}x\left(t\right)+b_{i}\right\}$ in (Equation 4). There exist several ways to implement fuzzy modeling [38,42,43]. Here, we chose the one described in [42,43]. Because each chaotic test system has nonlinear terms with a state variable in common, we only need two rules to model them in a fuzzy manner. Other chaotic systems may require a larger number of rules. For instance, in the transformed Rossler system, it is not possible to extract two linear terms from the premise variable [43], and therefore, more than two rules are required for the fuzzification of this system.

### 5.3. Fuzzy Modeling of Chaotic Systems

In this paper, we considered the fuzzy modeling of chaotic systems of integer order with hidden attractors, as listed in Table 1. Since the derivatives that appear in chaotic systems are of integer order, it is said that the chaotic systems are of integer order. The first column reports the name of the system and the author of the system. In the second column, the mathematical model is listed. In the third column, the parameters of the system are listed; in case these do not come from the seminal work, we cite the source from which we took them. In the last column, it is indicated if the chaotic system has equilibrium points. Below, we present the fuzzy modeling of each of the systems.

The chaotic system given by Chua [44,49] was one of the earliest to be investigated with hidden attractors [50]. This three-dimensional dynamical system is known to exhibit nonlinearities [49]. For the Chua system with hidden attractors, assume that $x_{1}\in \left[-d,d\right]$ and d>0. Then, the fuzzy model that exactly represents the chaotic system under $x_{1}\in \left[-d,d\right]$ is given by
(5)$$\begin{array}{c} \mathrm{Rule}\;1:\;\;\mathrm{IF}\;x_{1}\left(t\right)\;\mathrm{is}\;F_{1}\;\;\mathrm{THEN}\;\;\dot{x}\left(t\right)=A_{1}x\left(t\right)+b_{1}\\ \mathrm{Rule}\;2:\;\;\mathrm{IF}\;x_{1}\left(t\right)\;\mathrm{is}\;F_{2}\;\;\mathrm{THEN}\;\;\dot{x}\left(t\right)=A_{2}x\left(t\right)+b_{2} \end{array}$$
where $x={\left[x_{1},x_{2},x_{3}\right]}^{T}$, $b_{1}=b_{2}={\left[0,0,0\right]}^{T}$, d=1.1, $F_{1}\left(x_{1}\right)=\frac{1}{2}\left(1-\frac{\phi (x_{1})}{d}\right)$, $F_{2}\left(x_{1}\right)=1-F_{1}\left(x_{1}\right)$, and 
$$A_{1}=\left(\begin{array}{ccc} \alpha (d-1) & \alpha & 0\\ 1 & -1 & 1\\ 0 & -\beta & -\gamma \end{array}\right)\;\;\;\;\;A_{2}=\left(\begin{array}{ccc} -\alpha (d+1) & \alpha & 0\\ 1 & -1 & 1\\ 0 & -\beta & -\gamma \end{array}\right)$$
$$\phi \left(x_{1}\right)=\left\{\begin{array}{cc} \frac{\psi (x_{1})}{x_{1}} & \;\;\;x_{1}\neq \;0\\ m_{0} & \;\;\;x_{1}=0 \end{array}\right.\;\;\;\;\;\;\mathrm{sat}\left(x_{1}\right)=\left\{\begin{array}{cc} -1 & \;\mathrm{if}\;x_{1}\leq -1\\ x_{1} & \mathrm{if}\;|x_{1}|<1\\ 1 & \;\mathrm{if}\;x_{1}\geq 1 \end{array}\right.$$

The final output of the Chua fuzzy system is inferred by
$$\dot{x}=\sum_{i=1}^{2}F_{i}\left(x_{1}\left(t\right)\right)\left\{A_{i}x\left(t\right)+b_{i}\right\}.$$
The chaotic behavior of the Chua system around the hidden attractor is illustrated in Figure 1a.

Analogous to the Chua system, the Sprott A system assumes that $x_{2}\in \left[-d,d\right]$ and d>0, then the fuzzy model is
$$\begin{array}{c} \mathrm{Rule}\;1:\;\;\mathrm{IF}\;x_{2}\left(t\right)\;\mathrm{is}\;F_{1}\;\;\mathrm{THEN}\;\;\dot{x}\left(t\right)=A_{1}x\left(t\right)+b_{1}\\ \mathrm{Rule}\;2:\;\;\mathrm{IF}\;x_{2}\left(t\right)\;\mathrm{is}\;F_{2}\;\;\mathrm{THEN}\;\;\dot{x}\left(t\right)=A_{2}x\left(t\right)+b_{2} \end{array}$$
where $x={\left[x_{1},x_{2},x_{3}\right]}^{T}$, $b_{1}=b_{2}={\left[0,0,2\right]}^{T}$, d=5.0, $F_{1}\left(x_{2}\right)=\frac{1}{2}\left(1+\frac{x_{2}}{d}\right)$, $F_{2}\left(x_{2}\right)=\frac{1}{2}\left(1-\frac{x_{2}}{d}\right)$, and 
$$A_{1}=\left(\begin{array}{ccc} 0 & 1 & 0\\ -1 & 0 & d\\ 0 & -d & 0 \end{array}\right)\;\;\;\;\;A_{2}=\left(\begin{array}{ccc} 0 & 1 & 0\\ -1 & 0 & -d\\ 0 & d & 0 \end{array}\right)$$

The final output of the Sprott A fuzzy system is inferred by
$$\dot{x}=\sum_{i=1}^{2}F_{i}\left(x_{2}\left(t\right)\right)\left\{A_{i}x\left(t\right)+b_{i}\right\}$$
and the phase space of the Sprott A system is shown in Figure 1b.

For the NE_6_ system, assume that $x_{3}\in \left[-d,d\right]$ with d>0, then the fuzzy model is
$$\begin{array}{c} \mathrm{Rule}\;1:\;\;\mathrm{IF}\;x_{3}\left(t\right)\;\mathrm{is}\;F_{1}\;\;\mathrm{THEN}\;\;\dot{x}\left(t\right)=A_{1}x\left(t\right)+b_{1}\\ \mathrm{Rule}\;2:\;\;\mathrm{IF}\;x_{3}\left(t\right)\;\mathrm{is}\;F_{2}\;\;\mathrm{THEN}\;\;\dot{x}\left(t\right)=A_{2}x\left(t\right)+b_{2} \end{array}$$
where $x={\left[x_{1},x_{2},x_{3}\right]}^{T}$, $b_{1}=b_{2}={\left[0,0,-0.75\right]}^{T}$, d=3.0, $F_{1}\left(x_{3}\right)=\frac{1}{2}\left(1+\frac{x_{3}}{d}\right)$, $F_{2}\left(x_{3}\right)=\frac{1}{2}\left(1-\frac{x_{3}}{d}\right)$, and 
$$A_{1}=\left(\begin{array}{ccc} 0 & 1 & 0\\ 0 & 0 & 1\\ -d & -1-d & 0 \end{array}\right)\;\;\;\;\;A_{2}=\left(\begin{array}{ccc} 0 & 1 & 0\\ 0 & 0 & 1\\ d & -1+d & 0 \end{array}\right)$$

The final output of the NE_6_ fuzzy system is inferred by
$$\dot{x}=\sum_{i=1}^{2}F_{i}\left(x_{3}\left(t\right)\right)\left\{A_{i}x\left(t\right)+b_{i}\right\}$$
and the phase space of this system is shown in Figure 1c.

The LE_4_ system can be fuzzy modeled as follows. Assume that $x_{2}\in \left[-d,d\right]$ with d>0, then the fuzzy model is
$$\begin{array}{c} \mathrm{Rule}\;1:\;\;\mathrm{IF}\;x_{2}\left(t\right)\;\mathrm{is}\;F_{1}\;\;\mathrm{THEN}\;\;\dot{x}\left(t\right)=A_{1}x\left(t\right)+b_{1}\\ \mathrm{Rule}\;2:\;\;\mathrm{IF}\;x_{2}\left(t\right)\;\mathrm{is}\;F_{2}\;\;\mathrm{THEN}\;\;\dot{x}\left(t\right)=A_{2}x\left(t\right)+b_{2} \end{array}$$
where $x={\left[x_{1},x_{2},x_{3}\right]}^{T}$, $b_{1}=b_{2}={\left[0,0,0\right]}^{T}$, d=0.8, $F_{1}\left(x_{2}\right)=\frac{1}{2}\left(1+\frac{x_{2}}{d}\right)$, $F_{2}\left(x_{2}\right)=\frac{1}{2}\left(1-\frac{x_{2}}{d}\right)$, and 
$$A_{1}=\left(\begin{array}{ccc} 0 & 1 & 0\\ -1 & 0 & d\\ -d(a+b) & 0 & -d \end{array}\right)\;\;\;\;\;A_{2}=\left(\begin{array}{ccc} 0 & 1 & 0\\ -1 & 0 & -d\\ d(a+b) & 0 & d \end{array}\right)$$

The final output of the LE_4_ fuzzy system is inferred by
$$\dot{x}=\sum_{i=1}^{2}F_{i}\left(x_{2}\left(t\right)\right)\left\{A_{i}x\left(t\right)+b_{i}\right\}$$
and its phase space is shown in Figure 1d.

## 6. Fuzzy Synchronization

The general master–slave feedback control utilized to synchronize chaotic systems is summarized in Figure 2. In this paper, we used fuzzy control due to the characteristics that we explained in Section 1. The two fuzzy controls used to synchronize chaotic systems are described below.

### 6.1. Complete Fuzzy Synchronization

Let us consider the fuzzy representation of the chaotic system (Equation 2) as the master system and the system (Equation 6) as the slave system:(6)$$\begin{array}{c} \mathrm{Rule}\;1:\;\;\;\;\;\mathrm{IF}\;y_{1}\left(t\right)\;\mathrm{is}\;F_{1}\;\;\;\;\mathrm{THEN}\;\;\dot{y}\left(t\right)=A_{1}y\left(t\right)+b_{1}+Bu\left(t\right)\\ \mathrm{Rule}\;2:\;\;\;\;\;\mathrm{IF}\;y_{1}\left(t\right)\;\mathrm{is}\;F_{2}\;\;\;\;\mathrm{THEN}\;\;\dot{y}\left(t\right)=A_{2}y\left(t\right)+b_{2}+Bu\left(t\right) \end{array}$$
where $y={\left[y_{1},y_{2},y_{3}\right]}^{T}$ are state variables and $A_{1}$, $A_{2}$, $b_{1}$, $b_{2}$, $F_{1}$, $F_{2}$, and *d* have the same values as the master system (Equation 2). B is an input matrix, and u(t) is the control.

Defuzzification is given by
$$\begin{array}{c} \dot{y}=\sum_{i=1}^{2}\mu_{i}\left(y_{1}\left(t\right)\right)\left\{A_{i}y\left(t\right)+b_{i}\right\}+Bu\left(t\right),\;\mu_{i}\left(y_{1}\left(t\right)\right)=F_{i}\left(y_{1}\left(t\right)\right). \end{array}$$
The error is e(t)=y(t)−x(t). The error can be estimated from
(7)$$\begin{array}{c} \dot{e}\left(t\right)=\sum_{i=1}^{2}\mu_{i}\left(y_{1}\left(t\right)\right)\left\{A_{i}y\left(t\right)+b_{i}\right\}-\sum_{i=1}^{2}\mu_{i}\left(x_{1}\left(t\right)\right)\left\{A_{i}x\left(t\right)+b_{i}\right\}+Bu\left(t\right). \end{array}$$
The goal of the synchronization is to design the fuzzy control:(8)$$\begin{array}{c} u\left(t\right)=-\sum_{i=1}^{2}\mu_{i}\left(y_{1}\left(t\right)\right)\left\{C_{i}y\left(t\right)+b_{i}\right\}+\sum_{i=1}^{2}\mu_{i}\left(x_{1}\left(t\right)\right)\left\{C_{i}x\left(t\right)+b_{i}\right\} \end{array}$$
such that $∥e(t)∥\to 0$ when t→∞. Substituting the fuzzy control (Equation 8) in the error (Equation 7), one obtains
$$\begin{array}{c} \dot{e}\left(t\right)=\sum_{i=1}^{2}\mu_{i}\left(y_{1}\left(t\right)\right)\left\{\left(A_{i}-BC_{i}\right)y\left(t\right)\right\}-\sum_{i=1}^{2}\mu_{i}\left(x_{1}\left(t\right)\right)\left\{\left(A_{i}-BC_{i}\right)x\left(t\right)\right\} \end{array}$$
The gain matrices $C_{i}$ are calculated during the design process. With the idea of linearization given in [38,42], if there exists gain matrices such that
$$\begin{array}{c} {\left\{\left(A_{1}-BC_{1}\right)-\left(A_{2}-BC_{2}\right)\right\}}^{T}\;\times \left\{\left(A_{1}-BC_{1}\right)-\left(A_{2}-BC_{2}\right)\right\}=0 \end{array}$$
then the total error of the system becomes linear when $\dot{e}\left(t\right)=Ge\left(t\right)$ according to the fuzzy control (Equation 8), where $G=A_{1}-BC_{1}=A_{2}-BC_{2}$. Moreover, if G<0, the the error is asymptotically stable. This convergence was proven in [42].

### 6.2. Projective Fuzzy Synchronization

In the projective synchronization, the drive system is scaled to a scalar factor α. Let the master system (Equation 1) and the slave system be $\dot{y}=g\left(y\left(t\right),u\left(x\left(t\right),y\left(t\right)\right)\right)$. If there exist a constant α≠0 such that $\lim_{t\to \infty }∥y(t)-\alpha x(t)∥=0$, then the synchronization is referred to as projective, and α is called the *scaling factor*.

In particular, for the case study at hand, our master system is given by its T–S fuzzy modeling (Equation 9) and the slave system (Equation 11).
(9)$$\begin{array}{cc} \mathrm{Rule}\;1:\;\mathrm{IF}\;x_{1}\left(t\right)\;\mathrm{is}\;F_{1} & \;\mathrm{THEN}\;\dot{x}\left(t\right)=A_{1}x\left(t\right)\\ \mathrm{Rule}\;2:\;\mathrm{IF}\;x_{1}\left(t\right)\;\mathrm{is}\;F_{2} & \;\mathrm{THEN}\;\dot{x}\left(t\right)=A_{2}x\left(t\right) \end{array}$$

The final output of the fuzzy system (Equation 9) is inferred from
(10)$$\dot{x}=\sum_{i=1}^{2}\mu_{i}\left(x_{1}\left(t\right)\right)A_{i}x\left(t\right)$$
where μ satisfies the same conditions as (Equation 4).
(11)$$\begin{array}{cc} \mathrm{Rule}\;1:\;\mathrm{IF}\;x_{1}\left(t\right)\;\mathrm{is}\;F_{1} & \;\mathrm{THEN}\;\dot{y}\left(t\right)=A_{1}y\left(t\right)+u\left(t\right)\\ \mathrm{Rule}\;2:\;\mathrm{IF}\;x_{1}\left(t\right)\;\mathrm{is}\;F_{2} & \;\mathrm{THEN}\;\dot{y}\left(t\right)=A_{2}y\left(t\right)+u\left(t\right) \end{array}$$

The final output of the slave fuzzy system (Equation 11) is given by
(12)$$\dot{y}=\sum_{i=1}^{2}\mu_{i}\left(x_{1}\left(t\right)\right)\left\{A_{i}y\left(t\right)+u\left(t\right)\right\}$$

Let the error e(t)=y(t)−αx(t). Substituting the final outputs of the fuzzy systems (Equation 10) and (Equation 12), we obtain the derivative of the error:(13)$$\dot{e}\left(t\right)=\sum_{i=1}^{2}\mu_{i}\left(x_{1}\left(t\right)\right)\left\{A_{i}e\left(t\right)+u\left(t\right)\right\}$$

By means of PDC, which creates a controller fuzzy system with rule premises identical to those of the plant fuzzy system [51], the next fuzzy control is obtained:$$\begin{array}{cc} \mathrm{Rule}\;1:\;\;\;\;\;\;\mathrm{IF}\;x_{1}\left(t\right)\;\mathrm{is}\;F_{1}\;\mathrm{THEN} & \;u\left(t\right)=-\gamma A_{1}\left[y\left(t\right)-\alpha x\left(t\right)\right]-\gamma \left[y\left(t\right)-\alpha x\left(t\right)\right]\\ \mathrm{Rule}\;2:\;\;\;\;\;\;\mathrm{IF}\;x_{1}\left(t\right)\;\mathrm{is}\;F_{2}\;\mathrm{THEN} & \;u\left(t\right)=-\gamma A_{2}\left[y\left(t\right)-\alpha x\left(t\right)\right]-\gamma \left[y\left(t\right)-\alpha x\left(t\right)\right] \end{array}$$
where γ is a control parameter.

The final output of the fuzzy control is given by:
(14)$$\begin{array}{c} u\left(t\right)=\sum_{i=1}^{2}\mu_{i}\left(x_{1}\left(t\right)\right)\left\{-\gamma A_{i}\left[y\left(t\right)-\alpha x\left(t\right)\right]-\gamma \left[y\left(t\right)-\alpha x\left(t\right)\right]\right\} \end{array}$$

Substituting the fuzzy control (Equation 14) in the error (Equation 13), one obtains
(15)$$\dot{e}\left(t\right)=\sum_{i=1}^{2}\mu_{i}\left(x_{1}\left(t\right)\right)\left[\left(1-\gamma \right)A_{i}-\gamma I\right]e\left(t\right)$$
where I is the identity matrix.

A condition for the convergence of the error (Equation 15) is that, if there exists a positive definite symmetric constant matrix *P* and constant c>0, such that $\left[\left(1-\gamma \right)A_{i}-\gamma I\right){]}^{T}P+P\left[\left(1-\gamma \right)A_{i}-\gamma I\right)]\leq -cI$, i=1,2, then the error (Equation 15) is globally exponentially stable.

Another convergence condition for the error (Equation 15) concerns the characteristic values $\lambda_{j}$, j=1,…,n of the symmetric matrices $\left(1-\gamma \right)\left(A_{i}^{T}+A_{i}\right)-2\gamma I$, i=1,2. If every $\lambda_{j}$ has a strictly negative real part, then the error (Equation 15) is globally exponentially stable, which implies that (Equation 9) and (Equation 11) can asymptotically achieve projective synchronization. For the proof of these conditions, the reader is referred to the original work [52].

## 7. Projective Synchronizations with an Emphasis on Error Convergence

For projective synchronization, it is possible to find the value of the control parameter γ analytically so that the convergence of the synchronization error to zero can be made faster or slower according to the application. Through matrices $\left(1-\gamma \right)\left(A_{i}^{T}+A_{i}\right)-2\gamma I$, i=1,2, the characteristic values are found as a function of γ, and its range of variation is determined. Parameter γ is selected, ensuring the characteristic values have a strictly negative real part, and so, the second condition stated at the end of Section 6.2 is met. By means of the standard deviation of the characteristic values σ(λ), $\lambda =(\lambda_{1},\ldots ,\lambda_{n})$, found in the valid range of γ, the set of characteristic values that gives us the sought convergence (fast or slow). When σ(λ)=0, the set of associated characteristic values is very similar and not small in magnitude (due to the spectrum of variation of the values), and the convergence error quickly goes to zero. Then, determining the $\lambda^{*}=\arg\min\sigma \left(\lambda \right)=0$, in the range of variation of γ, the value of $\lambda^{*}$ is found so that the error converges as fast as possible. Otherwise, $\lambda^{*}=\arg\max\sigma \left(\lambda \right)$ is found.

The aforementioned procedure was carried out using known strategies to determine the maxima and minima of a scalar function with a vector argument. However, when the expressions as a function of γ are polynomials of a degree greater than 1, finding $\lambda^{*}$ is laborious in practice. In these cases (such as the ones discussed in this article), it is recommended to obtain the plot of γ vs. σ(λ(γ)) to determine the desired $\lambda^{*}$ and the range of variability of γ.

## 8. Simulation Results

In this section, we present the results of the numerical simulations of the chaotic systems with hidden attractors in Table 1 using the proposed fuzzy control approaches. We used the complete fuzzy and projective fuzzy synchronizations described in Section 6.1 and Section 6.2 to synchronize two identical chaotic systems with hidden attractors and different initial conditions. In the case of projective synchronization, we applied the strategy described in Section 7.

We further studied the rate of convergence by both control strategies applied to Chua’s system across a new set of initial conditions. The different initial conditions that we generated followed the methodology proposed by Danca [53]. With projective synchronization, we fixed the control parameter γ to the value that affords the fastest convergence, and we varied the scaling parameter and the different initial conditions. In each simulation, we looked for the first iteration from which each synchronization error was less than 2% for all subsequent iterations. With complete synchronization, we also varied the initial conditions and looked for the iterations that satisfy each synchronization error to be less than 2%. Finally, we compared the iterations found in order to analyze the convergence of the errors through them.

In addition, we analyzed different T–S fuzzy models and we found an effect of the fuzzy model of the tested chaotic system on the synchronization performance.

All the simulations were carried out on a personal computer with machine precision $2.2204\times 10^{-16}$ in Fortran with the GNU Fortran 10.3.0 compiler for Windows. No compiler options and optimizations were selected. The graphs were made with matplotlib 3.5.1.

In all numerical simulations, the differential equations were solved by using the Gautschi integration method [40] based on trigonometric polynomials. Unlike other fixed-step integration methods such as Euler or Runge–Kutta, the Gautschi method is a special integration method that takes the oscillations of chaotic systems into account. Gautschi’s method has been shown to exhibit accuracy closest to the variable-step method with control error ode45 and requires fewer evaluations of the chaotic system than Runge–Kutta 4 (RK4) [54,55].

In order to employ the Gautschi integration method, we ought to establish the system frequency and the integration step. If the system frequency is unknown, it is possible and advisable to underestimate it [40]. In the simulations carried out here, we used an underestimated frequency equal to 1 and an integration step of 1/128. With these values, we obtained the expected attractors, which we confirmed with the phase spaces of the test systems reported in the source articles. The simulation time was 150 time units for the systems Chua, NE_6_, and LE_4_ and 250 time units the Sprott A system. Despite their popularity, time units may be ambiguous, and we, therefore, report our simulations in terms of iterations instead. The total number of iterations was calculated as *(final time—initial time)/integration step*. In our case, 150 time units correspond to 19,200 iterations and 250 time units correspond to 32,000 iterations.

### 8.1. Results of Fuzzy Complete Synchronization

We synchronized each test system with another identical system, but with different initial conditions. First, we used the fuzzy models of each chaotic system with hidden attractors given in Section 5.3. Then, we applied the fuzzy control presented in Section 6.1.

The gain matrices $C_{1}$ and $C_{2}$ of each synchronization were obtained by solving the LMIs using the Matlab R2021b LMI toolbox (The MathWorks, Inc. Natick, MA, USA). Choose the input matrix B as the identity matrix.

Upon every synchronization, the partial errors converged to zero as the simulation progressed, that is all synchronizations were successful. The proposed control is robust and capable of synchronizing chaotic systems with hidden attractors.

#### 8.1.1. Chua

Two Chua systems with hidden attractors were synchronized under error-complete and initial conditions $x_{0}={\left[-3.7727,-1.3511,4.6657\right]}^{T}$ for the master system and $y_{0}={\left[0.01,0,0\right]}^{T}$ for the slave system [15]. In order to emphasize the difficulty of selecting a control to synchronize systems with hidden attractors, these systems were considered with the given initial conditions and feedback control to obtain synchronization failure [15].

To carry out the synchronization the gain matrices for the Chua chaotic system of integer order with hidden attractors with the chosen initial conditions were calculated. The matrices (Equation 16) were obtained. With these matrices, the error was asymptotically stable.
(16)$$C_{1}=\left(\begin{array}{ccc} 0.3364 & 1.1820 & 0\\ 1.1820 & -0.1250 & -1.3841\\ 0 & -1.3841 & 0.1237 \end{array}\right),\;\;\;\;C_{2}=\left(\begin{array}{ccc} -4.3145 & 1.1820 & 0\\ 1.1820 & -0.1250 & -1.3841\\ 0 & -1.3841 & 0.1237 \end{array}\right)$$

Substituting the gain matrices in the fuzzy control (Equation 8) and using the Gautschi integration method, we synchronized the two Chua chaotic systems of integer order with hidden attractors. The phase space is shown in Figure 3a. It can be appreciated that the slave system (dotted magenta line) successfully followed the master system (solid green line).

Figure 3b presents the evolution of the error in the three state variables in the full simulation. Leveraging the theoretical guarantees, since $∥e(t)∥\to 0$ when t→∞, it was concluded that the systems were successfully synchronized using the proposed fuzzy control plotted in Figure 3c. Further, all synchronization errors were lower than 2% from Iteration 12,301. This case asserts the feasibility of fuzzy control for synchronizing chaotic systems in the presence of hidden attractors. This fuzzy control overcomes the difficulties presented by chaotic systems with hidden attractors such as unexpected behaviors, sudden changes, or even system inoperability.

#### 8.1.2. Sprott A

Analogous to the previous synchronizations over the Chua system, we proceeded to synchronize the Sprott A system. The initial conditions used were $x_{0}={\left[0,5,0\right]}^{T}$ for the master system [56] and $y_{0}={\left[1.1,6.1,-0.1\right]}^{T}$ for the slave system. The calculated gain matrices for this system are
$$\begin{array}{cc} C_{1}=C_{2} & =\left(\begin{array}{ccc} 0.5 & 0 & 0\\ 0 & 0.5 & 0\\ 0 & 0 & 0.5 \end{array}\right) \end{array}$$

In Figure 3d, the phase space where the slave system (dotted line) mimics the master system (solid line) is shown. Figure 3e plots the three synchronization errors, all of which converged to 0. Furthermore, from Iteration 2792, all were less than 2%. Finally, Figure 3f depicts the behavior of the control in the three state variables. We thus concluded that the synchronization was successful.

#### 8.1.3. NE_6_

In the complete fuzzy synchronization of the NE_6_ system, we used as the initial conditions $x_{0}={\left[0,3,-0.1\right]}^{T}$ for the master system [47] and $y_{0}={\left[2.4,1.4,-0.6\right]}^{T}$ for the slave system. The gain matrices obtained with this system are
$$C_{1}=\left(\begin{array}{ccc} 0.5 & 0.5 & -1.5\\ 0.5 & 0.5 & -1.5\\ -1.5 & -1.5 & 0.5 \end{array}\right),\;\;\;\;C_{2}=\left(\begin{array}{ccc} 0.5 & 0.5 & 1.5\\ 0.5 & 0.5 & 1.5\\ 1.5 & 1.5 & 0.5 \end{array}\right)$$

Figure 3g shows the phase space, in which the slave system (dotted line) mimics the master system (solid line). Figure 3h plots the three synchronization errors, each converging to zero. From Iteration 1885, the three errors were less than 2%. In Figure 3i, the behavior of the fuzzy control used is presented. The controls $u_{1}$ and $u_{2}$ have the same behavior; hence, the graphs overlap. For all of the above, this synchronization was also successful.

#### 8.1.4. LE_4_

The initial conditions used to synchronize this system were $x_{0}={\left[0.2,0.7,0\right]}^{T}$ for the master system [48] and $y_{0}={\left[-0.1,0.3,0.1\right]}^{T}$ for the slave system. The gain matrices obtained are
$$C_{1}=\left(\begin{array}{ccc} 0.5 & 0 & -1.84\\ 0 & 0.5 & 0.4\\ -1.84 & 0.4 & -0.3 \end{array}\right),\;\;\;\;C_{2}=\left(\begin{array}{ccc} 0.5 & 0 & 1.84\\ 0 & 0.5 & -0.4\\ 1.84 & -0.4 & 1.3 \end{array}\right)$$

Figure 3j presents the phase space. The correct synchronization of the slave system (dotted line) with the master system (solid line) can be appreciated. Figure 3k shows the convergence of the errors to 0. From Iteration 3383, all errors were less than 2%. In Figure 3l, the behavior of the fuzzy control used is graphed. In this sense, synchronization was successful.

### 8.2. Results of Fuzzy Projective Synchronization

In this section, we present the results of the projective fuzzy synchronization for the test systems in Table 1. We considered the fuzzy models of the systems reported in Section 5.3. We applied the control in Section 6.2.

Fuzzy projective control (presented in Section 6.2) has the constraint that none of the chaotic systems involved, neither the master nor the slave, can have constant terms. Therefore, the only systems that can be synchronized with this control of those in Table 1 are Chua and LE_4_.

Here, we present the results of applying the strategy given in Section 7 to choose γ and that allows modulating the speed of convergence.

#### 8.2.1. Chua System

With those parameters presented in Section 5.3, we calculated the symmetric matrices $\left(1-\gamma \right)\left(A_{i}^{T}+A_{i}\right)-2\gamma I$, i=1,2, which are
$$\begin{array}{c} \left(\begin{array}{ccc} 1.69124(1-\gamma )-2\gamma & 9.4562(1-\gamma ) & 0\\ 9.4562(1-\gamma ) & -2(1-\gamma )-2\gamma & -11.0732(1-\gamma )\\ 0 & -11.0732(1-\gamma ) & 0.0104(1-\gamma )-2\gamma \end{array}\right) \end{array}$$
$$\begin{array}{c} \left(\begin{array}{ccc} -35.516(1-\gamma )-2\gamma & 9.4562(1-\gamma ) & 0\\ 9.4562(1-\gamma ) & -2(1-\gamma )-2\gamma & -11.0732(1-\gamma )\\ 0 & -11.0732(1-\gamma ) & -0.0104(1-\gamma )-2\gamma \end{array}\right) \end{array}$$

Subsequently, we looked for the characteristic values of each matrix to determine the range of values in which γ must lie to satisfy that the symmetric matrices have strictly negative characteristic values. The equation to find γ is nonlinear. Clearing γ analytically can be cumbersome. Hence, instead, we opted to plot the standard deviation of the characteristic values by varying the parameter γ in the interval [0.88,1] in Figure 4a. Such an interval can be found with trial and error with educated guesses; the sought γ values are those complying with the required convergence conditions.

Following the selection of the interval of the γ values that ensure convergence, we now present the synchronization under the lower and upper values of the function σ(λ(γ)) within the interval where the convergence requirements are met: γ=0.88 and γ=1.0, respectively, in the case of Chua. As the process involved some trial and error, we further show the synchronization for nearby values: γ=0.89 and γ=0.99. The same initial conditions used for the complete synchronization were used here: $x_{0}={\left[-3.7727,-1.3511,4.6657\right]}^{T}$ for the master system and $y_{0}={\left[0.01,0,0\right]}^{T}$ for the slave system [15]. The scaling factor used was α=0.5. The errors of each state variable of the synchronizations are plotted in Figure 5a–c. The error in the synchronizations with γ=0.99 and γ=1 converged to zero in fewer iterations than with γ=0.88 and γ=0.89 in each error. It can be appreciated how the value of γ affected the speed of convergence: faster with γ=0.99 and γ=1 and slower with γ=0.88 and γ=0.89, with implications for the domain application.

#### 8.2.2. LE_4_

When synchronizing this system projectively, we used the same initial conditions as for complete synchronization: $x_{0}={\left[0.2,0.7,0\right]}^{T}$ for the master system [48] and $y_{0}={\left[-0.1,0.3,0.1\right]}^{T}$ for the slave system. We chose the scaling factor α=0.5.

The symmetric matrices of this system are
$$\begin{array}{c} \left(\begin{array}{ccc} -2\gamma & 0 & -3.68(1-\gamma )\\ 0 & -2\gamma & 0.8(1-\gamma )\\ -3.68(1-\gamma ) & 0.8(1-\gamma ) & -1.6(1-\gamma )-2\gamma \end{array}\right) \end{array}$$
$$\begin{array}{c} \left(\begin{array}{ccc} -2\gamma & 0 & 3.68(1-\gamma )\\ 0 & -2\gamma & -0.8(1-\gamma )\\ 3.68(1-\gamma ) & -0.8(1-\gamma ) & 1.6(1-\gamma )-2\gamma \end{array}\right) \end{array}$$

Using again the strategy described in Section 7, when calculating the characteristic values of the matrices, we also found polynomials of the third degree, as happened with the Chua system. The plots γ vs. σ(λ(γ)) for γ∈[0.7,1.0] are shown in Figure 4b. Analogously, as we did with Chua, this interval was found by trial and error following educated guesses. The lower and upper values of the function where the convergence requirements were met occurred at γ=0.77 and γ=1.0, respectively. Again, because of the trial and error process, we also synchronized with nearby values: γ=0.78 and γ=0.99.

Figure 5d–f plot the error for each state variable, and again, we show how the value of γ affects the speed of convergence;: faster with γ=0.99 and γ=1.0 and slower with γ=0.70 and γ=0.71, in agreement with the strategy in Section 7.

### 8.3. A New Set of Initial Conditions

In this section, we analyze numerically the error convergence of both the complete and projective synchronizations of the Chua system with different initial conditions, and further, we investigate whether the scaling factor has an impact on the projective synchronizations.

To design further numerical simulations, like [53], first, we found the equilibrium points of the chaotic system; this system has three: two unstable [6.58831,0.0028364,−6.58547], [−6.58831,−0.0028364,6.58547] and one stable [0,0,0], plotted in Figure 1a. Then, we verified numerically the solutions that start from neighborhoods of the unstable equilibrium points, which are attracted to the point of stable equilibrium or tend to infinity. Finally, we visualize the hidden attractors by trial and error by choosing the starting points for the numerical simulation outside the neighborhoods of the unstable equilibrium points. The list of initial conditions used is listed in Appendix A. These simulations were also carried out using Gautschi integration with an underestimated frequency of 1 and an integration step of 1/128. Again, the simulated time spanned 150 time units, equivalent to 19,200 iterations per simulation.

Afterwards, we conducted both the complete and the projective synchronizations. For the case of complete synchronization, we also used the gain matrices given in (Equation 16). For the projective synchronization, we chose to simulate scaling factors α equal to 0.5, 1, 2, 3, 4, and 8. The control parameter was fixed to γ=0.99 in all projective simulations, which gave the fastest convergence, as estimated with the strategy given in Section 7.

In principle, the complete synchronization and the projective synchronization with α=1 are equivalent. However, the design of the respective control differed. We emphasize here such differences in our simulations.

All simulations were successful. Of course, some converged faster than others. To objectively establish this, we looked for the first iteration from which each synchronization error was less than 2% for each of the 30 newly found initial conditions. Figure 6 shows the box plot. Comparing the complete and projective synchronizations, there was a difference of 10,000 iterations between them, approximately, representing ∼50% of all 19,200 iterations.

A paired-samples right-tailed *t*-test was conducted to compare the results between the complete and the projective α=1 synchronizations. There was a significant difference between the scores for the complete (mean (μ) ± standard deviation (σ): μ±σ=13,573±2032.9) and projective (μ±σ=1641.2±256.85) conditions; t(29)=32.5046,p<0.001.

We conducted a repeated-measures ANOVA test (Matlab’s R2021b ranova) to analyze the effect of the scaling factor α on the six tested projective synchronizations α = 0.5, 1, 2, 3, 4, 8. We obtained F(5,140)=22.05, p<0.001. The post hoc comparison was conducted using a multiple comparison test (Matlab’s R2021b multcompare), where a Dunn and Sidak’s adjusted alpha revealed significant differences between α=0.5, and the remaining αs, p<0.05, as well as α=8 and the others αs, p<0.05. The anterior can be appreciated in Figure 6.

### 8.4. Alternative T–S Fuzzy Models

In the synchronizations presented so far, we used the T–S fuzzy models provided in Section 5.3. However, for a given dynamical system, the fuzzy model is not unique [42,43]. A proposal to systematically obtain other T–S fuzzy models is the tensor product (TP) model transformation [57,58,59]. In order to establish the effect of the fuzzy model on the performance of the synchronization, using TP model transformation [58], we obtained two further variants of the T–S fuzzy models from each test system and compared their performances. These fuzzy models are the sum normalization non-negativeness (SNNN) type and the close-to-normality (CNO) type. In this article, we call the T–S fuzzy models presented in Section 5.3 Base, the SNNN type SNNN, and the CNO type CNO.

All synchronizations with the different T–S fuzzy models were successful with the errors converging to zero. To objectively compare the simulations, we used the synchronization error and looked for the first iteration that satisfied that each error was less than 2% for all subsequent iterations.

The results of these simulations are reported in Table 2. The results showed that, with the alternative T–S fuzzy models in complete fuzzy synchronization, the performances of the simulations improved or were equal. With the Chua system, an improvement of 79.7% was obtained. For the Sprott A system, it was 4.6%. With the NE_6_ and LE_4_ systems, the performances of Base and the alternatives SNNN and CNO matched. In the case of projective fuzzy synchronization, this was only carried out with the Chua and LE_4_ systems due to the constraints of the fuzzy projective control, as explained in Section 8.2. With the Chua system, an improvement of 23.2% was obtained, while with the LE_4_ system, 8.5% more iterations were required with the alternative SNNN, and the results of the alternative CNO and Base coincided.

## 9. Discussion

In this research, multistable chaotic systems [60] with hidden attractors were considered. When the hidden attractors of the systems are unknown, unexpected behaviors could be more frequent. The challenge of synchronizing them can be blamed on the fact that, currently, there is no standard strategy for the identification of all possible attractors, thus escaping unwanted behaviors. Determining the basins of attraction is a daunting task and highly time-consuming in most cases. The fuzzy controls used in this work evidenced robustness in handling multistable systems with hidden attractors. Specifically, we studied systems without equilibrium points (Sprott A and NE_6_), with a line of equilibrium points (LE_4_) and with a stable equilibrium point (Chua). All the synchronization efforts were successful, even in the case of the Chua system, where the synchronization through feedback control must take special care in the selection of its coupling parameter to avoid leaving the hidden attractor [15].

The tested chaotic systems were chosen arbitrarily from the literature. These systems are interchangeable with any other continuous integer-order chaotic system with nonlinear terms with a state variable in common. Chaotic maps and continuous systems of fractional order were excluded in this study. Chaotic systems with delay or with unknown parameters and/or initial conditions were not considered either. These latter would require a different design for the fuzzy controls used.

Regarding the initial conditions, 30 new initial conditions for the Chua system were found and presented in Section 8.3 and Appendix A. The corresponding synchronizations were performed, evidencing that the fuzzy controls used are also robust to changes in the initial conditions.

In this work, the Gautschi special method of numerical integration was used. This method exploits the oscillatory characteristic of chaotic systems. This is a fixed-step method that can use integration steps up to $10^{-1}$ and resources almost comparable to Euler’s method, providing competitive results to a fixed-step method with error control [54,55]. Few works in the literature pay attention to this fundamental aspect in numerical simulations of chaotic systems, but even variable-step methods with error control can present problems [39].

In traditional controls, pole assignment is a standard strategy to maximize the convergence rate (of the control). However, in fuzzy projective synchronization, this strategy is unfeasible because the equations of the fuzzy control (Equation 14) and the dynamics of the error (Equation 15) do not allow moving the poles in a simple manner. Instead, the values of the gamma parameter that satisfy the convergence conditions are sought. Therefore, a new strategy was proposed to optimize the convergence of the error in the fuzzy projective synchronizations through the analysis of the control parameters statistics.

Every dynamical system has infinitely many fuzzy models that can approximate it. Considering this, we also showed that the synchronization performance was influenced by the choice of this model.

## 10. Conclusions and Future Work

We studied the feasibility of synchronizing chaotic dynamic systems with hidden attractors using fuzzy control. We exemplified the success of such an approach with a case study on the Chua system, a case where a previous attempt using a different control strategy was difficult. We did not intend to send the message that fuzzy control is somehow superior to other forms of controls, but instead, we just intended to highlight that, under some circumstances, this strategy might present specific advantages. For instance, to apply the fuzzy controls presented here, it suffices to fuzzify the system (analogously to Section 5.1 and Section 5.2) and then continue with Section 6 to calculate the gain matrices or the control parameter. Importantly, the control design in this manner does not depend on the initial conditions. This is in contrast to the effort by [16] on the coupling variant of the problem for this same system.

We showed the flexibility of the approach by designing controls over two different synchronization errors; complete and projective. We stated the conditions that provide theoretical guarantees for the convergence to the solution. These conditions were previously reported in the literature. Here, we went one step further by empirically showing the practical consequences.

For projective synchronization, we proposed a new strategy for error convergence optimization. This strategy capitalizes on the theoretical conditions of the control through the parameter γ. Consequently, the proposed strategy can be applied to any chaotic system that is synchronized with projective fuzzy control.

We focused on four case studies. However, when reviewing the literature, we did not find theoretical limitations for this approach to be applicable to other cases of chaotic systems with hidden attractors. Moreover, for our case studies, the results behaved as predicted by theory. Hence, we believe it may be possible to extend our fuzzy control strategy to other chaotic systems.

## Figures and Tables

**Figure 1 entropy-25-00495-f001:**
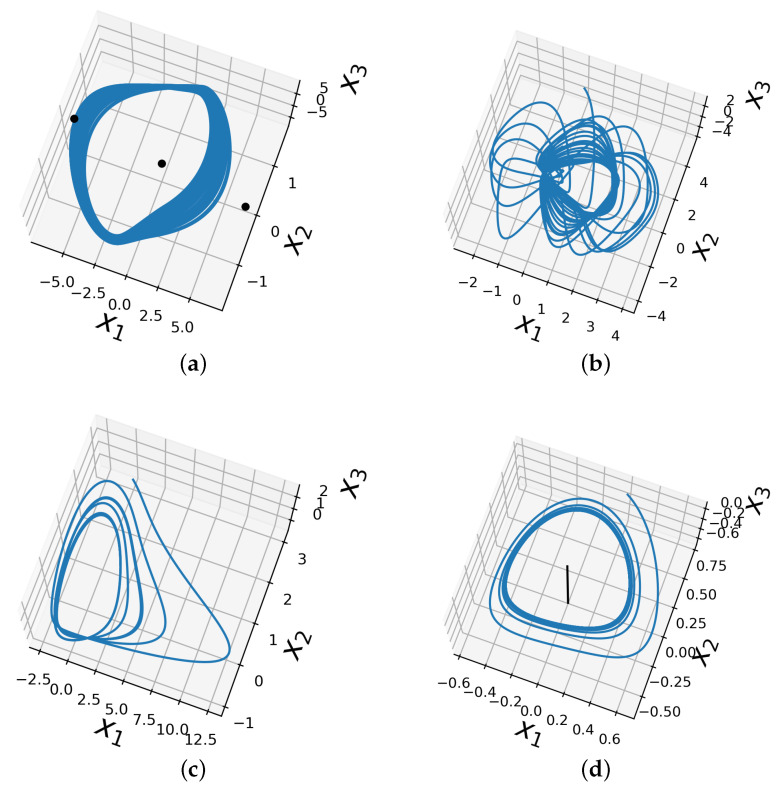
Hidden attractors of the test systems. (**a**) Chua system; the three equilibrium points are shown with black color to highlight the presence of the hidden attractor. (**b**) Sprott A system. (**c**) NE_6_ system. (**d**) LE_4_ system; the line of equilibrium $x_{3}$ has is plotted in black.

**Figure 2 entropy-25-00495-f002:**
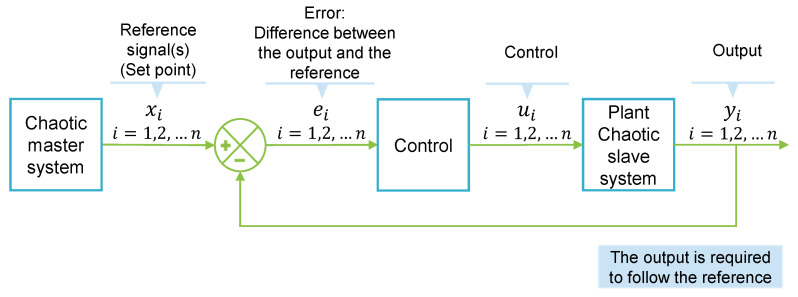
Synchronization scheme. Depiction of a master–slave feedback control where the output variables $y_{i}$ are required to follow the references (set points) generated by the master system $x_{i}$. The error $e_{i}$ is the difference between the output and the reference. The control $u_{i}$ forces the output to follow the reference.

**Figure 3 entropy-25-00495-f003:**
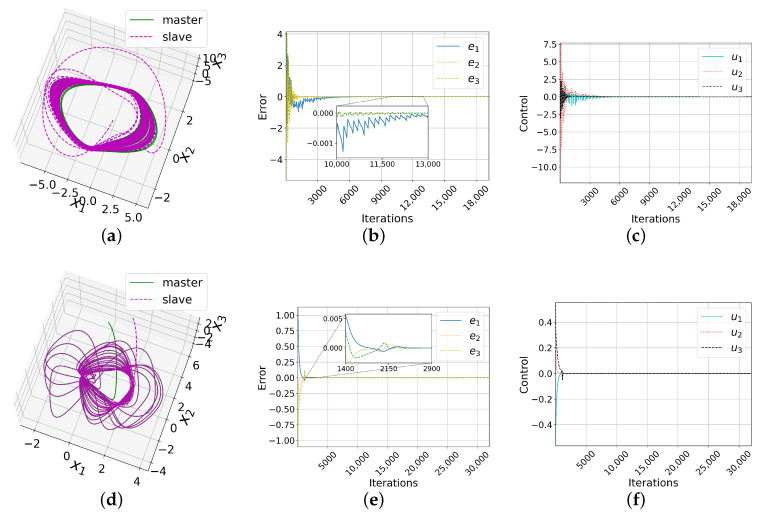
Results of complete synchronizations. (**a**) Phase space, (**b**) error, and (**c**) control of the Chua system. Analogously for the Sprott A system (**d**–**f**), NE_6_ system (**g**–**i**), and LE_4_ system (**j**–**l**). For Chua’s system, the synchronization errors were less than 2% from Iteration 12,301 onwards. For all other systems, this happened around Iteration 3000. If the units of time are seconds(s), 3000 iterations would be equivalent to 23.43 s, representing 15% of the total simulated time (150 s).

**Figure 4 entropy-25-00495-f004:**
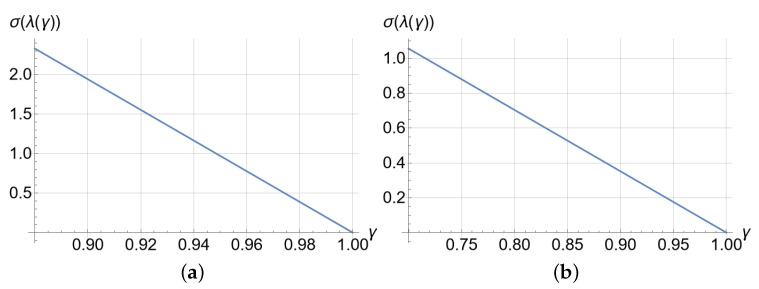
Standard deviation of the characteristics’ values. (**a**) Standard deviation of the Chua system. (**b**) Standard deviation of the LE_4_ system. Each one of the functions in its respective interval presents an absolute minimum and an absolute maximum.

**Figure 5 entropy-25-00495-f005:**
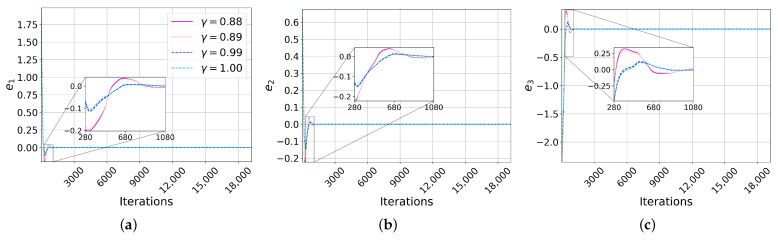
Errors of projective synchronizations. (**a**) Error in first state variable, (**b**) in the second variable, and (**c**) in the third variable of the Chua system. Analogously for the LE_4_ system (**d**–**f**). For the Chua system, the synchronizations with γ=0.99 and γ=1.00 reached zero faster than with γ=0.88 and γ=0.89. For the LE_4_ system, the fastest convergences were with γ=0.99 and γ=1.0 and slowest with γ=0.70 and γ=0.71. The range of variation for γ was [0.88,1] for Chua and [0.70,1] for LE_4_. The maximum difference between the synchronization errors was approximately 0.25 in Chua and 0.04 in LE_4_.

**Figure 6 entropy-25-00495-f006:**
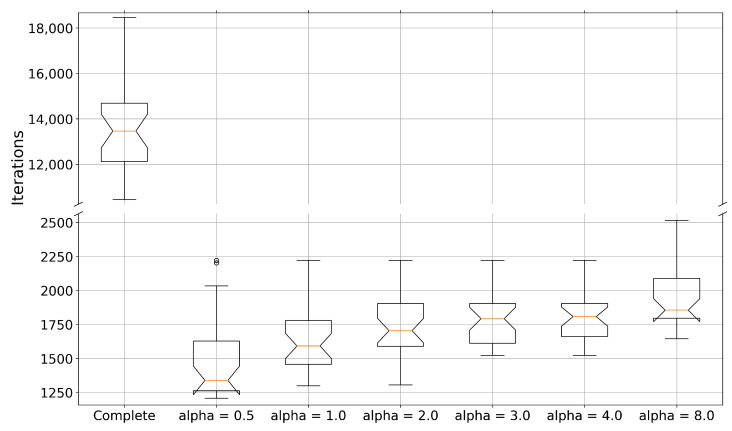
Iteration at which error synchronization is less than 2% for all 30 of the newly found initial conditions. While the complete and projective synchronizations with α=1 are theoretically analogous, note that the control design differs and, hence, the observed differences.

**Table 1 entropy-25-00495-t001:** Chaotic systems with hidden attractors. In the first column, the name of each system is presented. In the second column, the mathematical model is given. In the third column, the parameters of the system are cited if they do not come from the seminal work. The fourth column indicates whether the systems have equilibrium points.

System Name	Mathematical Model	Parameters	Eq. Point
Chua	${\dot{x}}_{1}=\alpha \left(x_{2}-x_{1}-\psi \left(x_{1}\right)\right)$	α=8.4562,	Yes
[44]	${\dot{x}}_{2}=x_{1}-x_{2}+x_{3}$	β=12.0732,	
	${\dot{x}}_{3}=-\left(\beta x_{2}+\gamma x_{3}\right)$	γ=0.0052,	
	$\psi \left(x_{1}\right)=m_{1}x_{1}+$	$m_{0}=-0.1768$,	
	$\left(m_{0}-m_{1}\right)$	$m_{1}=-1.1468$	
	$\mathrm{sat}(x_{1})$	[45]	
Sprott A	${\dot{x}}_{1}=x_{2}$		No
[46]	${\dot{x}}_{2}=-x_{1}+x_{2}x_{3}$		
	${\dot{x}}_{3}=1-x_{2}^{2}$		
NE_6_	${\dot{x}}_{1}=x_{2}$		No
[47]	${\dot{x}}_{2}=x_{3}$		
	${\dot{x}}_{3}=-x_{2}-x_{1}x_{3}-$		
	$x_{2}x_{3}-a$	a=0.75	
LE_4_	${\dot{x}}_{1}=x_{2}$		Line
[48]	${\dot{x}}_{2}=-x_{1}+x_{2}x_{3}$	a=4	
	${\dot{x}}_{3}=-ax_{1}x_{2}-bx_{1}x_{2}-$	b=0.6	
	$x_{2}x_{3}$		

**Table 2 entropy-25-00495-t002:** Performance comparison of fuzzy synchronizations using different Takagi–Sugeno (T–S) fuzzy models. The name of the chaotic system is in the first column. The T–S fuzzy models are in the second column: Base refers to the T–S fuzzy models given in Section 5.3. Sum normalization non-negativeness (SNNN) and close-to-normality (CNO) are the variants obtained with the transformation of the tensor product (TP) model. The first iterations in which the synchronization errors satisfied the given comparison criteria using complete and projective fuzzy control are reported in the third and fourth columns, respectively.

System	T–S Fuzzy Model	Complete	Projective
Chua	Base	12,301	1670
	SNNN	2496	1281
	CNO	2496	1281
Sprott A	Base	2792	
	SNNN	2663	
	CNO	2663	
NE_6_	Base	1885	
	SNNN	1885	
	CNO	1885	
LE_4_	Base	3383	1771
	SNNN	3383	1923
	CNO	3383	1771

## Data Availability

A free Fortran implementation of the synchronizations can be downloaded at https://github.com/Jessica-ZM/Fuzzy_Synchronizations#fuzzy_synchronizations.

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
