# Peer review of "Fuzzy Synchronization of Chaotic Systems with Hidden Attractors"

_entropy, 2023, doi:10.3390/e25030495_

Round 1

Reviewer 1 Report

Some references are redundant.

The section in line 235 is missing (replace ?? with the number).

Which are the potential applications of this fuzzy-based algorithm?

Which are the requirements of the chaotic system in order to support the fuzzy algorithm?

Why Fortran was used for simmulations?

Extended comments are attached.

Author Response

Please see the updated attachment.

Reviewer 2 Report

Dear Author

Happy New Year

You are initiating good job. Solving chaotic system via fuzzy controller.

Already plenty of papers did this, what is the novelty in this work?

Include your contribution and deliberate the work.

Thank you 

Author Response

Please see the updated attachment.

Reviewer 3 Report

The paper is fine, however there is a key critical point.

The whole analysis and design is based on the consequent system matrices.

The consequent system matrices depend on the TS fuzzy representation. However, there are infinite number of equivalent TS fuzzy models. Once we change the antecedent fuzzy sets the consequents will be changed as well.

Thus the same dynamic model has an infinite number of TS fuzzy models. Each alternative fuzzy model leads to different results of the design and analysis. The paper presents the antecedent fuzzy sets. There is no any explanation why this antecedents are used. Why not other antecedent?

Recent papers shows that many of design techniques are very sensitive for the shape of the antecedent fuzzy sets, hence the consequent matrices.

please study publications:

EEE Transaction: How to Vary the Input Space of a T–S Fuzzy Model: A TP Model Transformation-Based Approach, 

IEEE Transaction: „The Generalized TP Model Transformation for TS Fuzzy Model Manipulation and Generalized Stability Verification” 

Books „TP-Model Transformation-Based-Control Design Frameworks” and “Tensor Product model transformation in polytopic model based control”

Most recent special issue titled as “TP model transformation…..” Asian Journal of Control

I suggest to use TP model transformation to easily generate various TS fuzzy model alternatives. Espacially the SNNN type and CNO type (loose and tight convex hull defined by the vertices).

I also provide here a simple MATLAB code for general cases, but I also attach a file where the models given in the paper are substituted to create alternatives. I hope this may help the authors to check various alternatives TS fuzzy models.

Once the sensitivity  of the  design and the analysis to the alternatives are investigated

then the paper is completed.

Round 2

Reviewer 2 Report

Dear Authors,

Thank you for providing updated articles.

It looks good and will be interesting to read.

Best wishes

Reviewer 3 Report

The paper is a high quality work.

all my comments and critiques were taken seriously, with a comprehensive analysis around.